# Herbal Compounds Dauricine and Isoliensinine Impede SARS-CoV-2 Viral Entry

**DOI:** 10.3390/biomedicines11112914

**Published:** 2023-10-27

**Authors:** Shaneek Natoya Dabrell, Yi-Chuan Li, Hirohito Yamaguchi, Hsiao-Fan Chen, Mien-Chie Hung

**Affiliations:** 1International Master’s Program of Biomedical Sciences, China Medical University, Taichung 406040, Taiwan; u110206901@cmu.edu.tw; 2Department of Biological Science and Technology, China Medical University, Taichung 406040, Taiwan; ycl@mail.cmu.edu.tw; 3Graduate Institute of Biomedical Sciences, China Medical University, Taichung 406040, Taiwan; hyamaguchi@mail.cmu.edu.tw; 4Research Center for Cancer Biology, China Medical University, Taichung 406040, Taiwan; 5Center for Molecular Medicine, China Medical University Hospital, China Medical University, Taichung 406040, Taiwan; 6Cancer Biology and Precision Therapeutics Center, China Medical University, Taichung 406040, Taiwan; 7Institute of Biochemistry and Molecular Biology, China Medical University, Taichung 406040, Taiwan; 8Department of Biotechnology, Asia University, Taichung 413305, Taiwan

**Keywords:** SARS-CoV-2, vaccines, COVID-19, pandemic

## Abstract

Targeting viral entry has been the focal point for the last 3 years due to the continued threat posed by SARS-CoV-2. SARS-CoV-2’s entry is highly dependent on the interaction between the virus’s Spike protein and host receptors. The virus’s Spike protein is a key modulator of viral entry, allowing sequential cleavage of ACE2 at the S1/S2 and S2 sites, resulting in the amalgamation of membranes and subsequent entry of the virus. A Polybasic insertion (PRRAR) conveniently located at the S1/S2 site can also be cleaved by furin or by serine protease, TMPRSS2, at the cell surface. Since ACE2 and TMPRSS2 are conveniently located on the surface of host cells, targeting one or both receptors may inhibit receptor-ligand interaction. Here, we show that Dauricine and Isoliensinine, two commonly used herbal compounds, were capable of inhibiting SARS-CoV-2 viral entry by reducing Spike-ACE2 interaction but not suppressing TMPRSS2 protease activity. Further, our biological assays using pseudoviruses engineered to express Spike proteins of different variants revealed a reduction in infection rates following treatment with these compounds. The molecular modeling revealed an interconnection between R403 of Spike protein and both two compounds. Spike mutations at residue R403 are critical, and often utilized by ACE2 to gain cell access. Overall, our findings strongly suggest that Dauricine and Isoliensinine are effective in blocking Spike-ACE2 interaction and may serve as effective therapeutic agents for targeting SARS-CoV-2′s viral entry.

## 1. Introduction

The year 2020 onwards has been plagued with incredulity and consternation as the novel Coronavirus Disease 2019 (COVID-19) crippled economies and, more detrimentally, claimed the lives of approximately 7 million globally. In December 2020, vaccines and other therapeutic agents were approved to assist with minimizing the severe impact of the virus; however, neither approved therapeutic agents are capable of curtailing viral dispersion nor viral infection rates. Although three years have elapsed since this devastating virus emerged, much uncertainty remains, and COVID-19 is now documented as one of the deadliest pandemics in history, accounting for a massive loss of lives within the short period in which it has dominated. SARS-CoV-2 is the etiological agent responsible for the devastating COVID-19 outbreak that has been plaguing the world for the past three years. Coronaviruses date back to the 1960s when they first emerged. They are usually zoonotic in nature [1]; however, despite their zoonotic nature, they have evolved to engender human ailments [2].

Infection with SARS-CoV-2 is highly dependent on receptor-ligand interaction in a permissible and susceptible cell. Spike protein is a well-documented ligand essential for SARS-CoV-2 infection [3] and consists of two proteolytic cleavage sites, S1/S2, and S2′ to generate the subunits S1 and S2 that remain non-covalently linked [4]. Infection ensues when the S1 subunit of the Spike protein lodges onto the angiotensin-converting enzyme 2 (ACE2) receptor located on the surface of host cells [4], resulting in conformational changes signaling S2 to mediate membrane fusion. Viral entry is also permitted via proteolytic cleavage at the cell surface via transmembrane protease, serine 2 (TMPRSS2), or by furin at the polybasic insertion region (PRRAR) during cell trafficking [5]. ACE2 is a member of the angiotensin-converting enzyme family, which catalyzes the cleavage of Angiotensin I to angiotensin II and regulates the renin-angiotensin-aldosterone system (RAAS) [6]. Furthermore, ACE2 receptors are ubiquitous in tissues lining major organs of the body, particularly those of the lungs [7]. TMPRSS2 is also implicated in permitting viral entry as it is a well-documented ligand for the influenza virus [8,9] and SARS-CoV [10]. As a result, the ACE2 and TMPRSS2 receptors are key therapeutic targets for SARS-CoV-2.

Vaccinations developed for SARS-CoV-2 have been engineered to target viral entry, utilizing the virus’s Spike protein to do so [6]. Treatments with monoclonal and polyclonal antibodies are among the indications for severe COVID-19 infection; however, they are incapable of quelling viral entry. These indicated treatments are rather suited for targeting the symptoms associated with pathogenesis. Moreover, Meng et al. highlighted a significantly lowered vaccine efficacy over time, implying that vaccinations do not confer long-term protection [11].

Since its inception, the virus has mutated quite rapidly and, as a result, has maintained its dominance over society [12,13]. The quest to curtail COVID-19’s impact continues, as there is not a single FDA-approved treatment that can potentially prevent its spread or minimize its overall impact. Herbal medicine is an integral component of Traditional Chinese Medicine and was highly sought after to assist with minimizing the effects of the virus [14,15]. Dauricine and Isoliensinine are two common Traditional Chinese herbal compounds with numerous indications. Several reports have also outlined their significance in the field of research over the past years [7,16,17,18,19,20]. Dauricine, isolated from the Asian vine Menisperum Dauricum or Asian moonseed is used to treat a wide range of inflammatory diseases. It is also demonstrated that Dauricine could severe pneumonia co-infected with the influenza virus H5N1 and Streptococcus pneumoniae via the inhibition of NF-κB activation [19]. Isoliensinine, derived from Nelumbo Nucifera Gaertn, also known as lotus is indicated for the treatment of heart-related conditions in Chinese Herbal Medicine [18,20]. In addition, Isoliensinine was previously found to suppress Spike-ACE2 interaction and further inhibit SARS-CoV-2 infection [21,22].

In this study, we highlight the effects of naturally derived herbal derivatives on SARS-CoV-2’s viral entry. With the onset of the COVID-19 pandemic, there is an urgent need to establish a system for research in biosafety level 2 (BSL-2) facilities. Previously, researchers had developed a pseudotyped virus system for SARS-CoV and MERS-CoV, based on HIV-1 core, bearing the virus spike protein [23,24]. Pseudoviruses cannot undergo replication; therefore, they are harmless. The conformational structure of the surface proteins of pseudoviruses is highly correlated with the original virus. As such, pseudoviruses are designed to be a substitute for live viruses when evaluating viral entry inhibition via the substitution of their surface spike proteins with different variants of SARS-CoV-2. Further, the SARS-CoV-2 pseudovirus is also able to lower biosafety risks when compared to the native SARS-CoV-2 virus. Further, due to the presence of a luciferase gene, the pseudovirus system is quantifiable and highly reproducible. Therefore, considering safety and efficiency, we used this pseudovirus system in this study. Using a combination of biological and functional assays and VPP assays, we showed a decreased binding between the Spike protein and ACE2, which play a critical role in enabling SARS-CoV-2 viral entry. Altogether, we provide safe and effective herbal alternatives that may potentially curb surging infection rates associated with the highly mutating SARS-CoV-2 virus.

## 2. Methods and Materials

### 2.1. Herbal Compounds

All Herbal compounds were purchased from TargetMol (L6800, Chinese Pharmacopoeia Natural Product Library; TargetMol Chemicals Inc., Boston, MA, USA). Five hundred and fifty-eight (558) compounds were screened from the library prior to this study, leaving 34 suitable candidates. The cytotoxicity of these 34 compounds was then assessed using tetrazolium dye, 3-(4,5-dimethylthuazol 2-yl)-2,5-diphenylterazolium bromide (MTT) in 293T-ACE2 overexpressed cells.

### 2.2. Cell Culture

Vero E6 cells, which originate from African green monkey kidneys, are commonly used mammalian cell lines. Furthermore, they are extensively utilized when conducting coronavirus-related studies due to their high susceptibility to infection and viral replication rates [25]. Vero E6 cells were maintained in Dulbecco’s Modified Eagles Medium (DMEM), supplemented with 10% Fetal Bovine Serum (FBS), 1% penicillin/streptomycin (p/s), and 1× GlutaMAX. NCI-H460 cells, derived from lung adenocarcinoma, were maintained in Modified Eagle’s Medium (MEM), supplemented with 10% FBS and 1% p/s. 293T cells derived from human embryonic kidneys were maintained in DMEM, supplemented with 10% FBS and 1% p/s. 293T-ACE2 cell line was constructed by transfection of human ACE2 expressing plasmids (HG10108-UT; Sino Biological, Beijing, China) into 293T cells, followed by stable cell selection with hygromycin, and can be used for in vitro screening and characterization of drug candidates against SARS-CoV [26] and SARS-CoV-2 [27]. 293T-ACE2 cells were maintained in DMEM supplemented with 1% p/s, FBS, and 200 μg/mL hygromycin. All the cell lines used here were authenticated.

### 2.3. Cell Viability Assay

MTT assay was also used to assess cell viability in 293T-ACE2 overexpressed cells, NCI-H460 cells, and Vero E6 cells. 100 µL of ${1\times 10}^{4}$ 293T-ACE2 cells were seeded in 96 well plates and incubated overnight. 100 µL of ${0.5\times 10}^{4}$ cells were seeded for NCI-H460 and Vero E6, respectively. Following overnight incubation, cells were then treated with each compound for 24 h. A final concentration of 0.5 mg/mL of MTT reagent (M6494; Invitrogen, Carlsbad, CA, USA) was added and incubated for three hours. Results were corroborated at 570 nm using an ELISA reader. The least toxic compounds were further used for screening. For consistency and accuracy, experiments were conducted in triplicates.

### 2.4. Viral Pseudo Particle (VPP) Assay

Viral pseudoparticles (VPPs) were purchased from the RNAi core of Academia Sinica, Taiwan, and were used for this study. ${1\times 10}^{4}$ 293T-ACE2 and ${0.5\times 10}^{4}$ Vero E6 cells were seeded in 96 well plates and incubated overnight. For consistency and accuracy, experiments were performed in triplicates. After overnight incubation, cells were then pretreated with each compound and incubated for 1 h. Using a Multiplicity of infection (MOI) of 0.2, viral pseudoparticles were added and centrifuged at 1250 RPM at 4 °C for 30 min. The plates were then incubated for 24 h. CCK8 reagent (CK04; Dojindo Laboratories, Kumamoto, Japan) combined with DMEM was added, using a ratio of 1:4, and incubated for 3 h. Immediately following incubation, the luminescence was measured at 450 nm to assess cell viability. Following this, an equal volume of ready-to-use luciferase substrate (E2610, Bright-Glo Luciferase Assay System; Promega Corporation, Madison, WI, USA) was added, and luciferase activity levels were determined.

### 2.5. TMPRSS2 Protease Activity Assay

We first performed transient transfection by seeding 100 µL of ${2.5\times 10}^{4}$ 293T cells into 96 well plates. Plates were then incubated overnight to gain adequate confluency. Transfection of the TMPRSS2 plasmid was performed the following day. After 24 h, 90 µL of phosphate-buffered saline (PBS) containing the compounds was added to each well and pretreated for 1 h. The fluorogenic substrate Boc-QAR-AMC (ES014; R&D Biosystems, Minneapolis, MN, USA) was then added to each well, amounting to a final concentration of 100 µL. The fluorescence was measured (excitation 380 nm, emission 460 nm) using an ELISA reader after 1 h incubation at 37 °C.

### 2.6. Spike-ACE2 Interaction Assay

Interruption of the SARS-CoV-2 spike and human ACE2 interaction by different compounds was detected using TR-FRET assay according to the manufacturer’s protocol (79949; BPS bioscience, San Diego, CA, USA). ACE2 and spike proteins with and without different compounds were incubated at room temperature for 1 h. TR-FRET signals were measured at 665 nm with a 60-microsecond delay after laser excitation at 340 nm.

### 2.7. Western Blot Analysis

Western Blot was used to confirm the overexpression of ACE2 and TMPRSS2 in 293T cells. Cells were lysed using RIPA buffer. The supernatant was collected after centrifuge, which was used for analysis. Samples were separated on 10% SDS-PAGE and blotted on PVDF membrane. After blocking in 5% skim milk at room temperature for 1 h, the membrane was then incubated in the primary antibodies for 24 h, followed by the secondary antibody for 1 h. The membrane was washed with PBS 3 times, and the signals were visualized with an enhanced chemiluminescence (ECL) substrate. Primary antibodies used in this study were ACE2 (4355S; Cell Signaling Technology, Danvers, MA, USA), flag (MA1-91878; Invitrogen, Carlsbad, CA, USA), and β-Actin (C-2) (sc-8432; Santa Cruz Biotechnology, Dallas, TX, USA).

### 2.8. Statistical Analysis

Student’s *t*-test was used to determine the statistical significance of the data presented among different groups. A *p*-value of less than or equal to 0.05 is typically considered to be statistically significant.

### 2.9. Molecular Docking

The ability of both Dauricine and Isoliensinine to inhibit the interaction of the Spike protein of each variant and the hACE2 complex was assessed using iGEMDOCK [28]. The RBD (Omicron)-hACE2 complex (PDB ID: 7WBP) and the RBD (Delta)-hACE2 complex (PDB ID: 7WBQ) were used as the templates for the docking experiment. The 3D structure of both compounds was retrieved from the PubChem online database and used for further analysis. All docking tasks were performed using iGEMDOCK with the indicated parameters “Population size = 300, Generation = 80, and Number of solutions = 100”. The best pose was selected and used for further analysis in PyMOL [29]. The distances from the hACE2 complex and Spike protein to each compound were measured, and the common amino acid residue interactions were recorded. Binding affinities were then predicted using PRODIGY-LIG [30].

## 3. Results

### 3.1. Reduced Infection Rates Post-Drug Treatment with Dauricine and Isoliensinine

Initially, we sought to identify a safe and effective herbal compound that is potent enough to obstruct the interaction between Spike protein and cell surface receptors, ACE2 and TMPRSS2, which are instrumental in enabling SARS-CoV-2’s cell entry. Several compounds from the Chinese herbal medicine library (Figure 1) were screened using MTT assay as a corroborator in our 293T-ACE2 cell model (Figure 2A). Compounds with a toxicity rate of over 50% were eliminated, while others were subjected to infection using a VPP assay to assess their ability to reduce infection rates (Figure 2B). To further examine the efficiency of our VPP assay, we also used well-known inhibitors, hydroxychloroquine (HCQ) and oligomeric proanthocyanidins (OPCs), as positive controls (Appendix A) [31,32,33]. Several studies have reported that high expression levels of ACE2 are correlated with severe infection and COVID-19-related complications [34,35]. We therefore sought to understand whether ACE2 is indeed critical for SARS-CoV-2 infection. Using SARS-CoV-2 wild-type (WT) pseudovirus, we observed a significant increase in infection efficiency when ACE2 was overexpressed (Appendix A). This result confirmed that ACE2 is critical for infection with SARS-CoV-2 WT pseudovirus. In summary, these results highlighted a significant reduction in infection rates by compounds 10 (Dauricine) and 32 (Isoliensinine), suggesting that these two compounds play a role in reducing in vitro viral infection rates. 

### 3.2. Cytotoxicity of Dauricine and Isoliensinine

Since Dauricine and Isoliensinine were shown to be capable of reducing infection rates, we further assessed whether they affect cell viability. Dose-dependent cytotoxic assays were performed in human embryonic kidney-derived 293T cells, stably expressing ACE2 (Figure 3A,B), African green monkey kidney-derived VeroE6 cells (Figure 3C,D), and Lung adenocarcinoma derived NCI-H460 cells (Appendix A). Vero E6 cells have been widely used in many laboratories for SARS-CoV-like virus research and cell-culture-based infection models because of their high titers of viral replication. This high susceptibility could be related to the high expression level of ACE2 [36], which is used by both SARS-CoV and SARS-CoV-2 [37]. Because lung-related illnesses are consistent with the clinical manifestations of COVID-19, we also assessed the cytotoxicity of Dauricine and Isoliensinine in NCI-H460 cells. Cells were treated with different concentrations (3.125 μM, 6.25 μM, 12.5 μM, and 50 μM) of Dauricine and Isoliensinine, respectively, for 24 h and were then subjected to MTT assay. Notably, after treatment with the initial dose of 10 μM, cells maintained considerable viability, suggesting that the indicated concentration of 10 μM that was initially used to treat cells had no detrimental effect on cell viability. Our data further suggests that Isoliensinine (Figure 3B,D) has a less significant effect on cytotoxicity when compared to Dauricine (Figure 3A,C).

### 3.3. Antiviral Activities of Dauricine and Isoliensinine

Having confirmed the cytotoxic effect posed by both compounds, we investigated the potential of both compounds to attenuate infection rates in a dose-dependent manner. Using different concentrations of our compounds (0.08, 0.4, 2, and 10 μM), we pretreated cells for one hour, followed by infection with WT pseudoparticles. The data presented highlights an attenuation in infection rates in 293T-ACE2 (Figure 4A,B) and VeroE6 (Figure 4C,D) cells in a dose-dependent manner. Data analysis concluded that there was a more significant reduction in infection rates when treated with Isoliensinine (Figure 4B,D) as opposed to Dauricine (Figure 4A,C). Since both compounds were effective in reducing VPP infection rates following treatment, we then used the data to determine the safety window of both compounds in Table 1. We first determined the 50% Inhibitory Concentration (IC50) of both compounds by calculating the quantity of each compound required to reduce VPP infection rates by 50% using data obtained in Figure 4A–D. The IC50 Isoliensinine was significantly lower than that of Dauricine in both 293T-ACE2 and VeroE6 cells, suggesting that Isoliensinine had higher inhibitory potency against VPP infection. On the other hand, the 50% Cytotoxic Concentration (CC50), the concentration that reduces the number of viable cells by 50% compared with the control, was obtained from experiments in Figure 4A–D, and both values were then used to calculate the Selectivity Index (SI). The SI is a ratio that measures the window between cytotoxicity and antiviral activity by dividing the given IC50 value by the CC50 value (IC50/CC50). Theoretically, a compound with a higher SI ratio is considered safer and more effective for treatment. Together, these results suggested that both Dauricine and Isoliensinine are potent inhibitors of VPP infection without causing cytotoxicity. However, Isoliensinine’s higher SI index indicates higher safety when compared to Dauricine.

### 3.4. Dauricine and Isoliensinine Maintain Inhibitory Activities against Different Variants of VPPs

SARS-CoV-2 is a highly mutable virus with several variants. Each variant is disparate, possessing different numbers of mutations and utilizing different viral entry mechanisms, and each has a contrasting impact on host survival [16]. Its rapid mutations have been reported to occur in the receptor-binding domain of the Spike protein, where receptor interactions occur [16]. There have been documented instances of mutant strains of SARS-CoV-2 developing across several global regions. These include the B.1.1.7 variant detected in the United Kingdom in October 2020, the B.1.351 variant identified in South Africa in October 2020, the P.1 variant identified in Brazil in December 2020, the B.1.617 which emerged in India in 2021, accounting for a massive loss of lives due to its severity and high transmissibility and the highly transmissible Omicron variant (B.1.1.529) which emerged in the latter half of 2021, resulting in many infections and sub-mutations [38,39]. We therefore analyzed the ability of Dauricine and Isoliensinine to suppress infection rates post-treatment among the B.1.1.7, B.1.351, P.1, B.1.617 and the B.1.1.529 variants. We pretreated cells with the indicated concentration of both compounds, followed by infection with each VPP for 24 h and the later detection of luciferase activity. We observed similar inhibitory activity of Dauricine (Figure 5A,C) and Isoliensinine (Figure 5B,D) against the WT and other variants in 293T-ACE2 and VeroE6 cells. These results imply that Dauricine and Isoliensinine can maintain their potency for inhibiting viral entry of different SARS-CoV-2 variants.

### 3.5. Dauricine and Isoliensinine Obstruct Spike Protein- ACE2 Interaction but Not TMPRSS2 Protease Activity

Interaction between the Spike protein and cell surface receptors ACE2 and TMPRSS2 is of great significance as they permit the entry of the virus into permissible cells. A significant amount of emphasis has been placed on these pathways as they are the major targets for potential therapeutic agents. We reasoned that since Dauricine and Isoliensinine are able to reduce infection rates in vitro, they should have an impact on one or both cell surface receptors. First, we performed a FRET-based assay to investigate whether these two compounds affect the interaction between the Spike protein and ACE2. The data presented validates the mechanism utilized by both compounds, highlighting a reduced interaction between the Spike protein and the ACE2 receptor in the presence of Dauricine (Figure 6A) and Isoliensinine (Figure 6B). While our data suggests that Dauricine and Isoliensinine could limit this key receptor-ligand interaction, we then proceeded to evaluate the effect that these two compounds possess on TMPRSS2 protease activity. To this end, we performed a FRET-based assay to measure cellular TMPRSS2 protease activity. The data revealed that neither Dauricine nor Isoliensinine possessed any notable effect on TMPRSS2 protease activity (Figure 6C,D) when TMPRSS2 was overexpressed in 293T cells (Appendix A). To further examine the effects of Dauricine and Isoliensinine on TMPRSS2 protease activity, we also used well-known TMPRSS2 inhibitors, tannic acid and oligomeric proanthocyanidins (OPCs), as positive controls (Appendix A) [32,40]. To further analyze the effects of Dauricine and Isoliensinine on infection rates in the presence or absence of TMPRSS2, human TMPRSS2 and a control vector were ectopically expressed in 293T-ACE2 and VeroE6 cell lines (Appendix A). The results showed that for both Dauricine (Figure 7A,C) and Isoliensinine (Figure 7B,D), cells expressing TMPRSS2 were more resistant to the treatment. These results further support our claim that Dauricine and Isoliensinine lack the potency to inhibit TMPRSS2 protease activity in Figure 6C,D. In conclusion, these results suggested that the mechanism of Dauricine and Isoliensinine against SARS-CoV-2 viral entry is highly correlated with the interruption of ACE2-Spike interaction but not TMPRSS2 protease activity.

### 3.6. Molecular Modeling of Dauricine and Isoliensinine

Our findings in Figure 6A,B highlighted that our compounds can sufficiently reduce Spike-ACE2 interaction. To further detect binding sites associated with the reduction in such interaction, we performed molecular docking using the Spike protein receptor binding domain (RBD) of SARS-CoV-2 and its variants as the targets. The 3D structure of Dauricine and Isoliensinine were obtained from PubChem, and the binding poses were predicted via iGEMDOCK. The best pose of each compound was selected for further analysis in PyMOL. The interactions between WT/variants and compounds are listed in Appendix A, providing the potential ability to inhibit the Spike protein residues from ACE2 interaction. Our model shows that Dauricine tends to inhibit the Omicron variant (B.1.1.529) (Figure 8A). In contrast, Isoliensinine prefers to associate with the Omicron variant (B.1.1.529) and Delta variant (B.1.617) (Figure 8B,C). The binding energy predicted using PRODIGY-LIG was listed in Table 2, showing similar ΔG were generated during Omicron-Dauricine, Omicron-Isoliensinine, and Delta-Isoliensinine interactions. The surface electrostatic potential map with the best docking pose indicated that the environment of the binding pockets all are mainly hydrophobic (Figure 8D). Notably, the R403 residue of Spike protein possesses high availability for both Dauricine and Isoliensinine-binding, together with nearby polar residue N417 that contributes hydrogen bonds to the compounds. The critical role of R403 residue in ligand-protein interaction was also reported in a traditional Chinese medicine study [41], where the single-point mutation R403A resulted in a reduced binding affinity to Ginkgolic acid GA171. Together, we provided a binding model of two common herbal compounds, Isoliensinine and Dauricine, to the Spike protein. These results further support our model, suggesting that these herbal compounds may potentially inhibit the SARS-CoV-2 Omicron and Delta variants infection by disrupting Spike protein-ACE2 interaction.

## 4. Discussion

The last three years have been far from quintessential as a result of the emergence of COVID-19. The virus continues to linger today due to its rapid mutations, resulting in increasing infection rates and continued disruption of livelihood for those affected. Though not as severe as when it initially emerged, much uncertainty remains as treatment options are aimed at minimizing the risks associated with infection but fail to bring its spread to a halt. Furthermore, studies conducted on vaccinations show vaccines confer minimal protection as they fail to neutralize some variants [42,43]. Additionally, waning immunity can be observed among several variants following two doses of vaccination, with neutralization being rescued only via the administration of a booster [11]. Taking these factors into consideration, expanding treatment options for COVID-19 is imperative.

Taking into account that viral entry is highly dependent on the ACE2 and TMPRSS2 receptors, we conducted ACE2 overexpression in 293T cells and relied heavily on these cell models throughout our study. Due to numerous constraints associated with conducting live virus experiments, we employed the use of pseudoviruses for infection purposes. Pseudoviruses are safe and effective tools used for conducting virus-related studies without a large number of restraints. Unlike live viruses, which can replicate numerous times, pseudoviruses can only replicate once, making them suitable for BSL2 laboratories. There are several advantages associated with using pseudoviruses. The structure of SARS-CoV-2 pseudovirus is highly correlated with the original strain. Their similarities lie mainly in their surface protein structure, which allows them to remain effective in their ability to enter cells. Due to the presence of a luciferase gene, pseudovirus infections are easily detected, and experiments are highly reproducible. Psuedoviruses used in our study were specifically engineered using a VSV lentivirus vector system to express full-length S protein sequences of spike proteins associated with different variants.

Dauricine and Isoliensinine are two herbal compounds with numerous indications, including the treatment of autoimmune and heart disorders. One study provided evidence that Dauricine has the potential to effectively treat pneumonia cases that are co-infected with the influenza virus H5N1 and Streptococcus pneumoniae. This therapeutic effect was achieved via the inhibition of NF-κB activation, as indicated by previous research [19]. Furthermore, earlier studies have demonstrated the efficacy of Isoliensinine in inhibiting the interaction between Spike protein and ACE2 receptor, hence showing potential effectiveness against SARS-CoV-2 [21,22]. In our study, we used a biological approach. Findings from our VPP assay highlighted the inhibitory activity of Dauricine and Isoliensinine against viral entry of SARS-CoV-2 through interfering with the interaction of Spike and ACE2 protein. SARS-CoV-2 has a notable degree of genetic variability, resulting in the emergence of multiple distinct variants. These mutations have made it rather perplexed to successfully avert its impact. Furthermore, it has been shown that mutations in the furin-cleavage site region increase S1/S2 cleavage, which is highly correlated with viral pathogenesis [5,44]. Mutations occur in the receptor-binding domain of the Spike protein, which is responsible for receptor interaction. Therefore, we further investigated whether Dauricine and Isoliensinine retained their inhibitory effects against different variants. We observed that both compounds retained inhibitory effects, not only on the wild type but also on different variants of SARS-CoV-2.Most variants share similar mutations except the Omicron (B.1.1.529) and Delta (B.1.617) variants, which contain the E484A and L452R mutations, respectively. Interestingly, these two mutations are not found in other variants [11], suggesting that the E484A found in the Omicron variant and the L452R found in the Beta variant account for their destructive nature [45,46].

A previous study has reported that the Spike proteins of different variants are correlated with different viral entry pathways [11]. Furthermore, sequential cleavage of the Spike protein at the S1/S2 and S2′ sites is required for viral entry [6]. The S1 site interacts with the ACE2 receptor, which triggers the S2 site to mediate membrane fusion. Cellular protease TMPRSS2, also located on the cell surface, mediates cell entry via proteolytic cleavage of the S1/S2 site [13,44]. To further understand the viral entry mechanism utilized by both compounds, we tested their ability to reduce Spike protein interaction with cell surface receptors ACE2 and TMPRSS2. We found that both Dauricine and Isoliensinine are capable of reducing ACE2-Spike protein interaction but have no significant effect on TMPRSS2 activity. To further confirm that this phenomenon is also reflected in their ability to suppress infection, we used TMPRSS2 overexpression cells and found that the inhibitory activity of Dauricine and Isoliensinine declined in the TMPRSS2-expressing cells. Together, these findings indicated that both Dauricine and Isoliensinine were only involved in the interruption of Spike and ACE2 binding but did not affect the protease activity of TMPRSS2.

We docked the protein binding domain of each variant to ascertain the effect that our compounds possess on reducing the interaction of the Spike protein of several variants of SARS-CoV-2 to the ACE2 complex. Several reports have shown that the Spike protein of each variant is disparate, which ultimately contributes to the varying impact that each variant pose [11,47]. Notably, the differences in mutations among each variant ultimately affect infectivity and the mechanism of viral entry [11]. The Spike protein of the Delta (B.1.617) variant was previously shown to confer more efficient cell–cell fusion kinetics when compared to the WT [48,49] and showed more efficient syncytia formation, which was previously found to be associated with pathogenesis [50]. Interestingly, the R403 residue is pivotal for ACE2 and the Spike protein interaction. Zech et al. highlighted the role of Spike residue 403 on the binding of coronavirus spike proteins to human ACE2. RaTG13, a bat sarbecovirus, is a close relative of SARS-CoV-2, sharing 96% homology. In their published study, they show that a positively charged amino acid at position 403 is critical for the efficient utilization of ACE2 by the Spike proteins of bat coronaviruses [51]. Coincidentally, in our study, the R403 residue present contributes to an increased binding affinity for Dauricine and Isoliensinine. Logically, a higher increased binding affinity to both compounds further increases the chances of reducing Spike-ACE2 interaction. Moreover, hydrogen bonds are crucial in enabling protein-ligand interaction. We found that, in coordination with the R403 residue, the N407 polar residue contributes hydrogen bonds to the compounds. Pragmatically, the interactions of the N407 polar residue of the Spike protein and the R403 residue can constitutively promote a reduced interaction of the virus’s Spike protein and the ACE2 receptor. The Omicron variant (B.1.1.529) and the Delta (B.1.617) variant’s ability to reduce this crucial interaction can be attributed to the fact that the B.1.617 and, more significantly, the B.1.1.529 variant is known to have a higher affinity for ACE2 when compared to TMPRSS2 [11]. The Omicron variant also possesses a rather significant amount of mutations in comparison to Delta and other variants, most of which are located in the S1/S2 cleavage site. This is believed to further increase its affinity for ACE2, which it heavily relies on for viral entry [11].

## 5. Conclusions

Although there has been a significant reduction in the risk presented by COVID-19, the variability of SARS-CoV-2 creates obscurities, posing a continued threat to global health. The global prevalence of COVID-19 has positioned it as the primary cause of mortality and hospitalizations in many countries, underscoring the formidable characteristics of this viral pathogen and emphasizing the ongoing necessity for therapeutic interventions. In this study, we demonstrated the inhibitory effects of Dauricine and Isoliensinine on viral entry of SARS-CoV-2. Specifically, these compounds reduced the interaction between the Spike protein and the ACE2 receptor, impeding viral entry. However, it is important to note that they did not exhibit any suppressive effects on the activity of the TMPRSS2 protease. Moreover, our biological assays utilizing pseudoviruses engineered to express Spike proteins of various variants also revealed a decrease in infection rates after treatment with these compounds. Furthermore, the interactions of these compounds with the N417 polar residue and the R403 residue of the Spike protein may facilitate a reduction in the interaction between the Spike protein and the ACE2 receptor. Altogether, we propose Isoliensinine and Dauricine to be suitable candidates for the treatment of COVID-19.

## Figures and Tables

**Figure 1 biomedicines-11-02914-f001:**
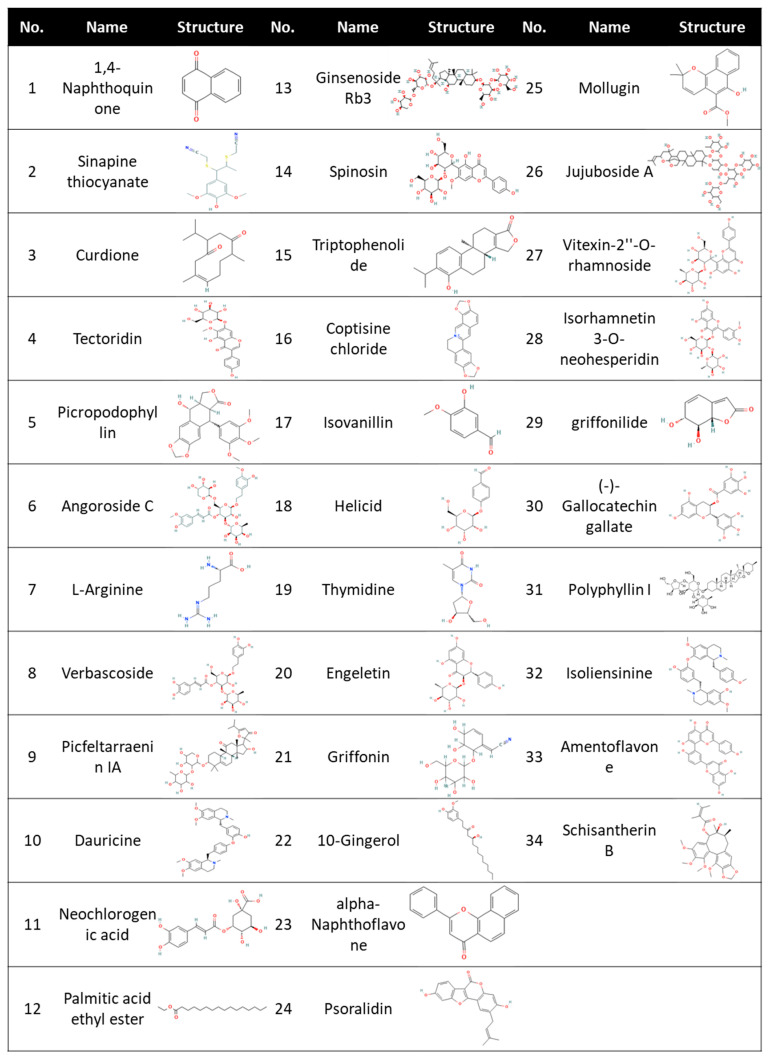
34 compounds screened from the Chinese Herbal Medicine Library. Chemical structure for Polyphyllin I was obtained from Medchemexpress, while all other chemical structures were obtained from PubChem.

**Figure 2 biomedicines-11-02914-f002:**
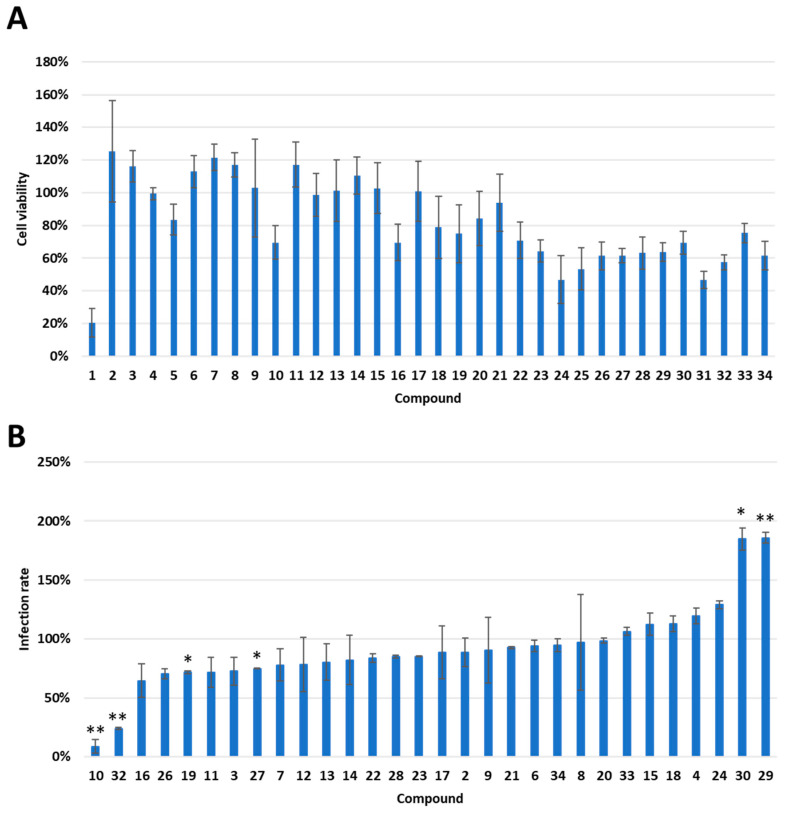
Screening of Chinese Herbal Medicine compounds reveals a reduction in SARS-CoV-2 wild-type (WT) Viral pseudoparticle (VPP) infection in the 293T-ACE2 cell line. (**A**) Cell viability was assessed following 24 h treatment with 10 µM of indicated compound or vehicle control (DMSO) in 293T-ACE2 cells by MTT assay. Values are normalized to vehicle control (100%) and shown as mean ± SD (n = 3). All data are shown as mean ± SD (*n* = 3). (**B**) 293T-ACE2 cells were pretreated with 10 μM of indicated compound or vehicle control (DMSO) for one hour and infected with SARS-CoV-2 WT-VPP. After 24 h of infection, the infection efficiency rate was measured according to luciferase activities. Values are normalized to vehicle control (100%) and shown as mean ± SD (*n* = 3). * *p* ≤ 0.05; ** *p* ≤ 0.01 compared to vehicle control.

**Figure 3 biomedicines-11-02914-f003:**
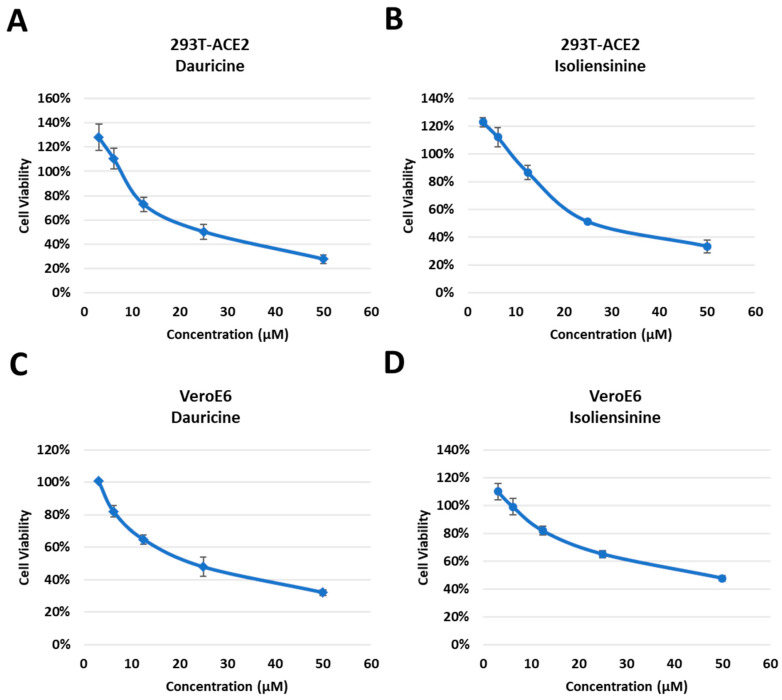
Cytotoxic activity of Dauricine and Isoliensinine in 293T-ACE2 and VeroE6 cells. (**A**,**B**) 293T-ACE2 cells were treated with different concentrations (3.125, 6.25 12.5, 25, and 50 µM) of Dauricine (**A**) or Isoliensinine (**B**), and cell viability was detected using MTT assay. Values are normalized to vehicle control (100%) and shown as mean ± SD (*n* = 3). (**C**,**D**) VeroE6 cells were treated with different concentrations (3.125, 6.25, 12.5, 25, and 50 µM) of Dauricine (**C**) or Isoliensinine (**D**), and cell viability was detected using MTT assay. Values are normalized to vehicle control (100%) and shown as mean ± SD (*n* = 3).

**Figure 4 biomedicines-11-02914-f004:**
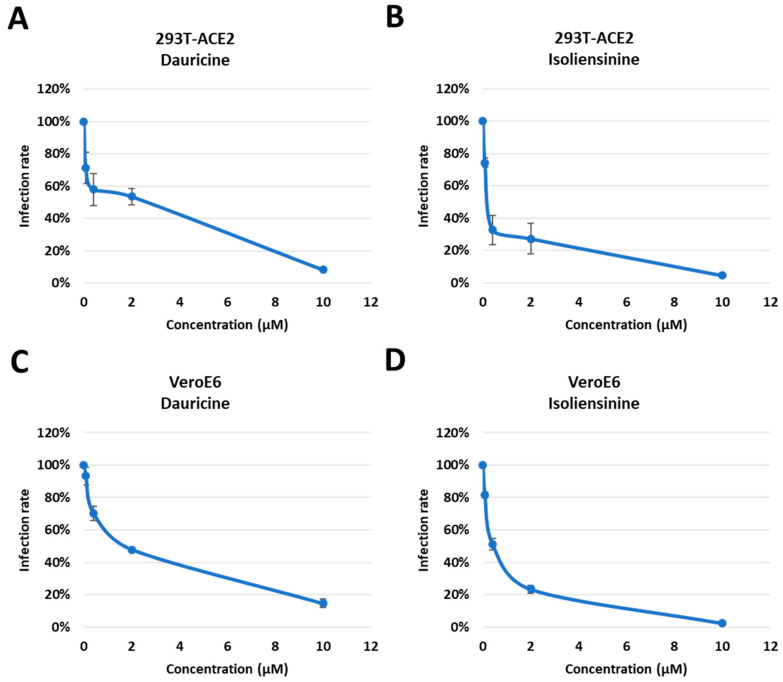
Trend in infection rates post-treatment with Dauricine and Isoliensinine. (**A**,**B**) 293T-ACE2 cells were pretreated with varying concentrations (0.08, 0.4, 2, and 50 µM) of Dauricine (**A**) or Isoliensinine (**B**) for one hour and infected with SARS-CoV-2 WT-VPP. After 24 h of infection, the infection efficiency rate was measured according to luciferase activities. Values are normalized to vehicle control (100%) and shown as mean ± SD (*n* = 3). (**C**,**D**) VeroE6 cells were pretreated with varying concentrations (0.08, 0.4, 2, and 50 µM) of Dauricine (**C**) or Isoliensinine (**D**) for one hour and infected with SARS-CoV-2 WT-VPP. After 24 h of infection, the infection efficiency rate was measured according to luciferase activities. Values are normalized to vehicle control (100%) and shown as mean ± SD (*n* = 3).

**Figure 5 biomedicines-11-02914-f005:**
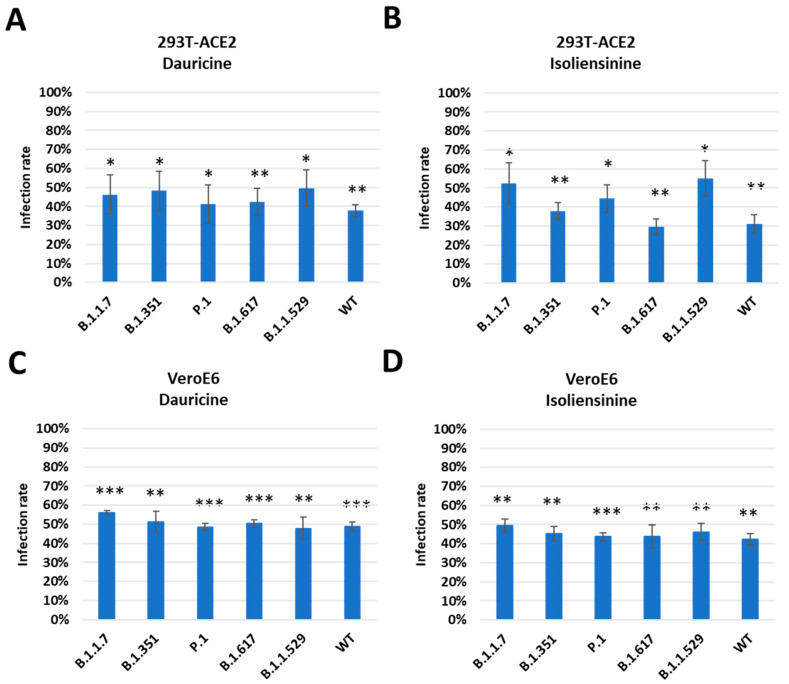
Trend in infection rates among mutant forms of SARS-CoV-2. (**A**,**B**) 293T-ACE2 cells were pretreated with 2 µM Dauricine (**A**) or 0.2 µM Isoliensinine (**B**) for one hour and infected with SARS-CoV-2 VPP of different variants. After 24 h of infection, the infection efficiency rate was measured according to luciferase activities. Values are normalized to vehicle control (100%) and shown as mean ± SD (*n* = 3). * *p* ≤ 0.05; ** *p* ≤ 0.01 compared to vehicle control. (**C**,**D**) VeroE6 cells were pretreated with 1.5 µM Dauricine (**C**) or 0.5 µM Isoliensinine (**D**) for one hour and infected with SARS-CoV-2 VPP of different variants. After 24 h of infection, the infection efficiency rate was measured according to luciferase activities. Values are normalized to vehicle control (100%) and shown as mean ± SD (*n* = 3). ** *p* ≤ 0.01; *** *p* ≤ 0.001 compared to vehicle control.

**Figure 6 biomedicines-11-02914-f006:**
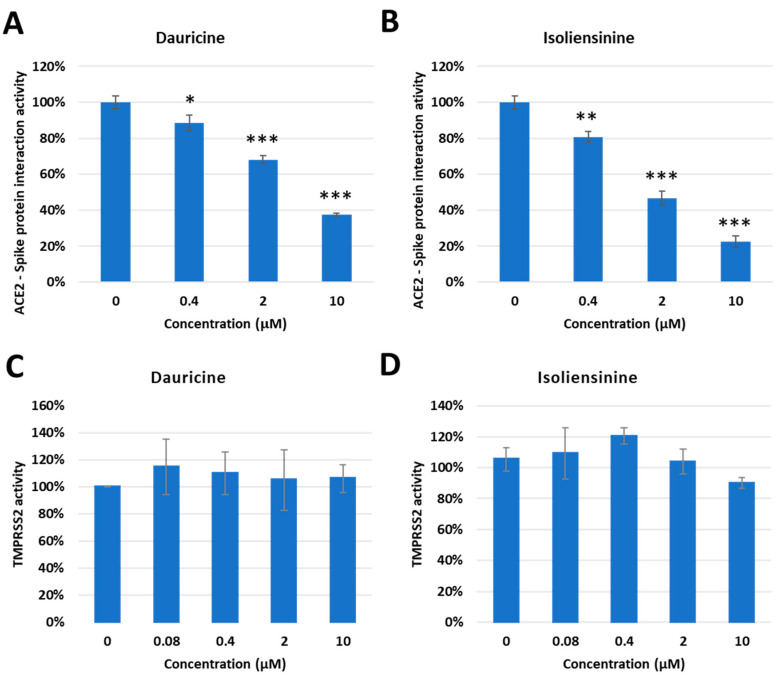
Inhibitory Effect of Dauricine and Isoliensinine on ACE2-Spike protein interaction and TMPRSS2 activity. (**A**,**B**) The percentage of Spike-ACE2 interaction from the FRET-base assay was shown with the indicated concentration of Dauricine (**A**) or Isoliensinine (**B**). Values are normalized to vehicle control (100%) and shown as mean ± SD (*n* = 3). * *p* ≤ 0.05; ** *p* ≤ 0.01; *** *p* ≤ 0.001 compared to vehicle control. (**C**,**D**) The TMPRSS2 enzymatic activity in vivo was measured by using a FRET-base assay with an increasing amount of Dauricine (**C**) or Isoliensinine (**D**). Values are normalized to vehicle control (100%) and shown as mean ± SD (*n* = 3).

**Figure 7 biomedicines-11-02914-f007:**
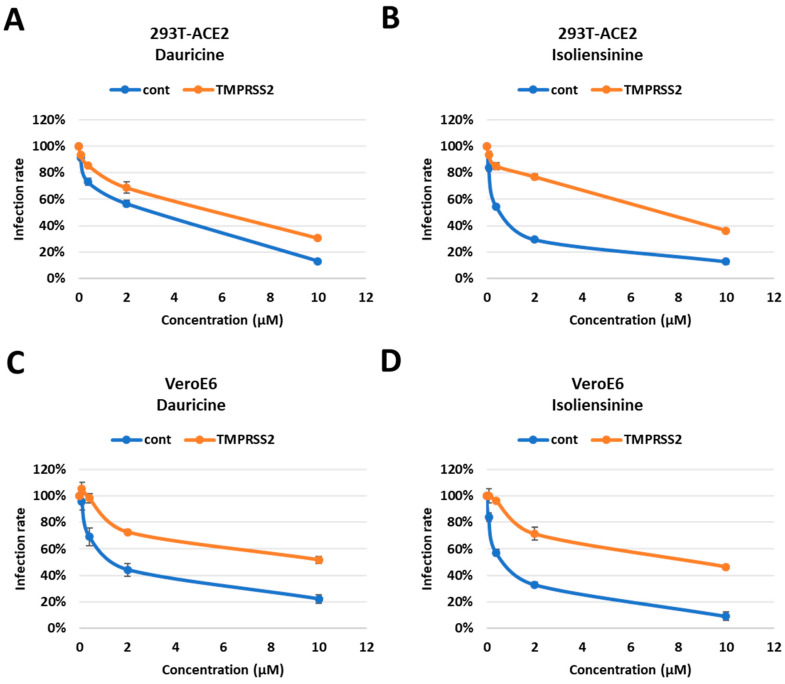
Impaired attenuation of infection rates in TMPRSS2 overexpressed cells. (**A**,**B**) 293T-ACE2 cells with and without TMPRSS2 expression were pretreated with the indicated concentrations (0.08, 0.4, 2, and 10 µM) of Dauricine (**A**) or Isoliensinine (**B**) and then infected with SARS-CoV-2 WT-VPP. After 24 h of infection, the infection efficiency rate was measured according to luciferase activities. Blue line (cont): empty vector control group; Orange line (TMPRSS2): TMPRSS2 overexpressing group. Values are normalized to vehicle control (100%) and shown as mean ± SD (*n* = 3). (**C**,**D**) VeroE6 cells with and without TMPRSS2 expression were pretreated with the indicated concentrations (0.08, 0.4, 2, and 10 µM) of Dauricine (**C**) or Isoliensinine (**D**), and then infected with SARS-CoV-2 WT-VPP. After 24 h of infection, the infection efficiency rate was measured according to luciferase activities. Blue line (cont): empty vector control group; Orange line (TMPRSS2): TMPRSS2 overexpressing group. Values are normalized to vehicle control (100%) and shown as mean ± SD (*n* = 3).

**Figure 8 biomedicines-11-02914-f008:**
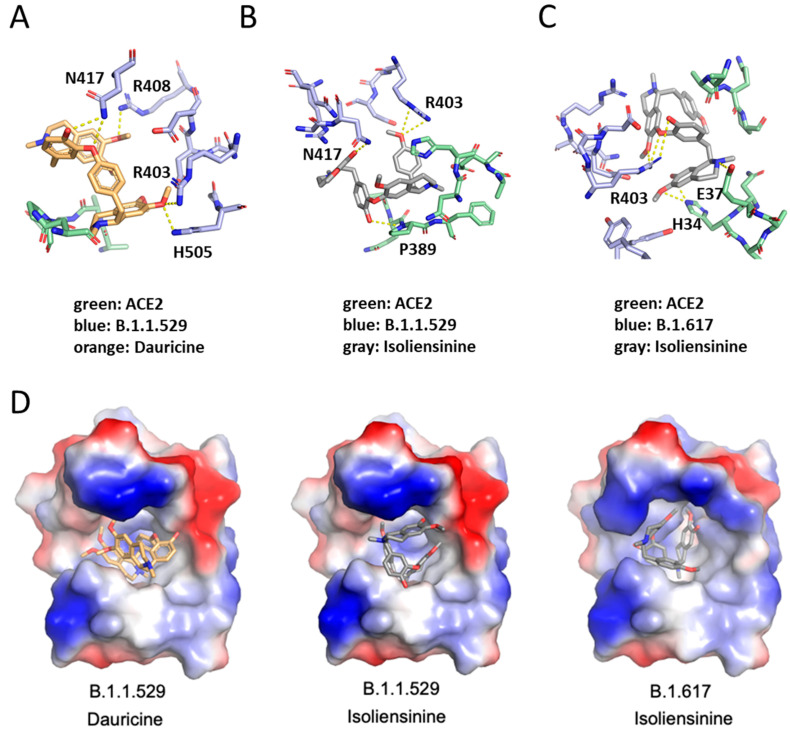
Molecular docking interaction of Dauricine and Isoliensinine in the binding pocket of SARS-CoV-2_RDB-hACE2 complex. The blue colors represent different variants of SARS-CoV-2_RDB residues, whereas green colors represent human ACE2. The orange colors represent Dauricine, whereas the gray colors represent Isoliensinine. (**A**) The binding mode between Dauricine and B.1.1.529 variant of SARS-CoV-2_RDB-hACE2 complex. (**B**) The binding mode between Isoliensinine and B.1.1.529 variant of SARS-CoV-2_RDB-hACE2 complex. (**C**) The binding mode between Isoliensinine and B.1.617 variant of SARS-CoV-2_RDB-hACE2 complex. (**D**) The surface electrostatic potential map with the best docking pose. Dauricine and Isoliensinine are colored orange and gray, respectively. Electrostatic surface potentials are colored red and blue for negative and positive charges, respectively.

**Table 1 biomedicines-11-02914-t001:** Cytotoxicity and antiviral activity of Dauricine and Isoliensinine against SARS-CoV-2 VPP based on Figure 2 and Figure 3. ^a^ IC50 is the half-maximal effective concentration such as the concentration of a compound required to inhibit SARS-CoV-2 infection by 50%. Values are shown as mean ± SD (*n* = 3). ^b^ CC50 is the median cytotoxic concentration, such as the dose causing 50% cell death. Values are shown as mean ± SD (*n* = 3). ^c^ SI is the selectivity index, such as the ratio of CC50 to EC50 (SI = CC50/IC50).

Cell Line	Compound	^a^ IC50 (μM)	^b^ CC50 (μM)	^c^ SI
293T-ACE2	Dauricine	1.26 ± 0.5	39.79 ± 7.59	31.83
	Isoliensinine	0.29 ± 0.11	43.58 ± 0.94	150.28
VeroE6	Dauricine	1.47 ± 0.18	25.36 ± 0.72	17.25
	Isoliensinine	0.45 ± 0.04	54.13 ± 5.9	120.29

**Table 2 biomedicines-11-02914-t002:** Binding affinities corroborated via protein binding energy prediction (PRODIGY).

	ΔG Prediction (Kcal/mol)
B.1.1.529 and Dauricine	−4.96
B.1.1.529 and Isoliensinine	−5.04
B.1.617 and Isoliensinine	−4.95

## Data Availability

The authors declare that all data supporting the findings of this study are available from the corresponding author upon reasonable request.

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
