# Peer review of "Herbal Compounds Dauricine and Isoliensinine Impede SARS-CoV-2 Viral Entry"

_biomedicines, 2023, doi:10.3390/biomedicines11112914_

Round 1

Reviewer 1 Report (Previous Reviewer 4)

The article is well written with some interesting findings. 

The Introduction describes the significance of the research. 

Additionally, the results are presented in a well-defined way. 

However, I strongly suggest writing the conclusion more scientifically. 

Mention the major findings of the study. Moreover, it can be linked with the future directions. 

Best Wishes  

The English language is acceptable. However, the final spell checks are required. 

Author Response

Reviewer 2 Report (Previous Reviewer 3)

The authors of the manuscript biomedicines-2654942 replied to the Reviewer’s suggestions and improved the quality of the manuscript. Therefore, I recommend the publication of the work.

Author Response

Reviewer 3 Report (Previous Reviewer 2)

The manuscript by Dabrell and colleagues shows important improvements compared to the previous version. However, some important points should be addressed before accepting it for publication:

1. The previous literature data must be properly cited and commented. Dauricine was already tested against infectious diseases, and this must be explicated in the main text (ref. 19). More seriously, the authors neither mention nor comment on the fact that Isoliensinine was already screened against SARS-CoV-2 infection (https://doi.org/10.1038/s41392-020-00343-z), and its biological activity characterized for RBD-ACE binding impairment, and whole virus infection neutralization (https://doi.org/10.1016/j.phrs.2021.105820).

2. Neutralization experiments were performed using VPP of the different VOCs. Please specify the advantage of using such a platform in the entry inhibition assay compared to a whole virus infection assay.

Minor points:

Figure 4: Please delete from the graph DMSO histograms, if Dauricine and Isoliensine data were normalized to DMSO control.

Figure 6: Please specify what "cont" stands for.

Table 2: Specify if IC50 ± SD is expressed, or other statistical analysis (± CI?)

Table captions must be placed before the tables, and supplementary data must be placed after the reference list

Materials and methods: add catalog number and distributor of the reagents, especially antibodies (i.e. mAbs used in the WB)

Author Response

Reviewer 4 Report (Previous Reviewer 1)

1. There had published papers on Dauricine and Isoliensinine. What is the new finding on your study and previous study?

2. The author used 293 T cells and Vero E6 cells. Did you try in TMPRSS2 cell line?

3. Please describe IC50 and CC50 of those 2 compounds in the text.

4. why the author didnot use live virus in this paper? Please discuss it in discussion part.

5. The author said those compounds affects on viral entry on ACE 2 receptor of 293 T cells and Vero E 6 cells. How about other cell lines which expressed ACE 2 receptor.

6. why the author used VPP assay? VPP is based on which virus variants?

7. Fro antiviral effect, the author used LUC assay. Why the author didnot try using real time PCR,  plaque Assay to confirm viral inhibition ?

Round 2

Reviewer 3 Report (Previous Reviewer 2)

The authors addressed all my comments.

This manuscript is a resubmission of an earlier submission. The following is a list of the peer review reports and author responses from that submission.

Round 1

Reviewer 1 Report

The study described dauricine and isoliensinine delay SARs-cOV-2 entry. I have some quries on manuscript.

1. The author used Viral Psudo particle as virus. why the author donot use live virus and explain it in introduction.

2. what are the advantages of viral Psudo particle. please describe it in discussion part.

3. The infection rate reduction is calculated by luciferase asssay. Why the author donot use other methods for viral detection (eg- real time PCR, Plaque Assay)?

4. The author used Vero E6 , Calu-3, ACE2 expressing 293T cell. I found that Calu 3 cell only used for cell viability assay. Why Calu 3 cell donot use anymore in other assays? please write it in discussion part.

5. All figures are low resolution and quality. The X and Y axis in all figures should be more clear.

6. SARS-COV-2 variants also used Psudo Viral particles? Explain it why you used it in discussion part.

7. Herbal 2 compounds are effect on ACE 2 receptor and not TMPRSS-2 cells.

8. Those herbal drugs already used in other viral diseases. If so, please discuss it in discussion part.

Reviewer 2 Report

The manuscript by Dabrell et al. characterizes the effect of two promising natural compounds against SARS-CoV-2 infection. This result would be significant for the scientific community if the experiments had been performed and described with accuracy, unfortunately this is not the case.

In the abstract, introduction and discussion parts there are serious conceptual errors:
- spike protein is the only protein involved in the virus entry. It has both furin and tmprrs2 cleavage sites, and are processed by both proteases. if tmprss2 is not present on the surface of the target cell, then catepsins are involved in the entry process. Thus, two entry pathways has been widely described in the literature, and the authors should report this aspect in the introduction.
- s protein is comprised of two subunits (S1 and S2), they are not associated to the s protein.
- the description of the SARS-CoV-2 genome in the introduction is not useful for explaining the mechanism of action of the compounds, on the contrary, I interrupted the logical thread.
- The different viral variants are mutated throughout the genome, but the classification has been made con sidering the S sequence only. Moreover, the mutations are definitely not originated because of "viral particles circulating in the atmosphere"
- the authors compared the COVID-19 pandemics to others responsible of deaths 10 and 100 times higher, some statements should be toned down.

In the materials and methods section different cell lines are reported, and their ATCC reference or any other should be reported. Moreover:
- their names should be the same throughout the manuscript (e.g. CALU3 or Calu3)
- VeroE6 cells have precise molecular characteristics for which they are used in viral studies, and the authors must make this explicit
- Hek 293/ACE cells: it is not clear if they have been generated by others or by the authors, please explain better the protocol and cloning strategy used or add the proper reference.

Last, the results part lacks of the proper experimental controls to support the authors' hypothesis:
- what do the 293/ACE cells experiments add to the results obtained with VeroE6? Based on what has been said above for the importance of tmprss2, it would have been useful for at least one of the two lines to be engineered with this receptor.
- Thewe evaluation of the effect of the two compounds on the tmprss2 viral entry pathway it was not done, precisely due to the lack of an engineered cell line to have expressed it, and comparing the results with the same line without tmprrss2.
- Figure 4 should be changed showing all the dilutions tested to obtain a reliable IC50. If it hasn't been done, it is incorrect to define those concentrations as IC50.

The authors use little-used words and logical forms. The reading is unnecessarily convoluted and difficult to follow the logical thread. The language used is sensationalist and unscientific.

Reviewer 3 Report

The manuscript biomedicines-2448621 reports the biological evaluation of natural compounds. Since the authors did not provide any positive controls in the biological assays, I do not recommend the publication of this work. Please, find below my main comments:

1) Please, provide the chemical structure in the Table of Figure 1A. This is important for the reader to identify the main class of the assayed compounds.

2) Please, provide the standard deviation for the IC50 and EC50 values.

3) The authors did not use controls in the cell-based assays. What is the reason? Please, provide at least two positive controls for all assays.

4) Please, improve the number of points in Figure 3.

5) The authors provide Tables as Figures. Please, correct it.

6) It is not clear the reason that the authors used three cell lineages. The Calu-3 cells are the best model to evaluate potential inhibitors of SARS-CoV-2. Please, see some examples: 10.3389/fmicb.2022.997539, 10.1128/spectrum.00774-21, 10.3390/v13122434

7) Please, provide a chemical-structure explanation for isoliensinine and dauricine are best candidates than the other evaluated natural products.

8) For in silico calculations provide the protein surface and electrostatic potential map for the best docking pose. Additionally, provide the distance and main interactive mode of each amino acid residue with each inhibitor (as Table).

...

Reviewer 4 Report

The artic entitled “Herbal Compounds Dauricine and Isoliensinine impede SARS-CoV-2 viral entry” is an important research amid the ongoing evolution of SARS-CoV-2.

The study has been conducted in a well mannered way, and the article is scientifically sound. My concern is about the studies incorporated in the manuscript. I suggest to incorporate some recent related articles in the manuscript.
Add the following after 35 reference, 

https://doi.org/10.1016/j.biopha.2022.113522

Write COVID-19 and SARS-CoV-2 like this throughout the manuscript.

Overall the manuscript Is well  written.